# Heating Drinking Water in Cold Season Improves Growth Performance via Enhancing Antioxidant Capacity and Rumen Fermentation Function of Beef Cattle

**DOI:** 10.3390/antiox12081492

**Published:** 2023-07-26

**Authors:** Tengfei He, Shenfei Long, Guang Yi, Xilin Wang, Jiangong Li, Zhenlong Wu, Yao Guo, Fang Sun, Jijun Liu, Zhaohui Chen

**Affiliations:** 1College of Animal Science and Technology, China Agricultural University, Beijing 100193, China; tenghe@ethz.ch (T.H.); longshenfei@cau.edu.cn (S.L.); sy20203040712@cau.edu.cn (G.Y.); sy20213040719@cau.edu.cn (X.W.); jli153@cau.edu.cn (J.L.); wuzhenlong@cau.edu.cn (Z.W.); guoyaocau@163.com (Y.G.); liujijun@cau.edu.cn (J.L.); 2State Key Laboratory of Animal Nutrition and Feeding, Beijing 100193, China; 3Institute of Animal Huabandry, Hei Longjiang Academy of Agricultural Sciences, Harbin 150086, China; hljxmsf@163.com

**Keywords:** water temperature, cold season, growth performance, antioxidant capacity, rumen function, beef cattle

## Abstract

The research aimed to investigate the suitable drinking water temperature in winter and its effect on the growth performance, antioxidant capacity, and rumen fermentation function of beef cattle. A total of 40 beef cattle (640 ± 19.2 kg) were randomly divided into five treatments with eight cattle in each treatment raised in one pen according to initial body weight. Each treatment differed only in the temperature of drinking water, including the room-temperature water and four different heat water groups named RTW, HW_1, HW_2, HW_3, and HW_4. The measured water temperatures were 4.39 ± 2.546 °C, 10.6 ± 1.29 °C, 18.6 ± 1.52 °C, 26.3 ± 1.70 °C, and 32.5 ± 2.62 °C, respectively. The average daily gain (ADG) showed a significant linear increase during d 0 to 60 and a quadratic increase during d 31 to 60 with rising water temperature (*p* < 0.05), and the highest ADG of 1.1911 kg/d was calculated at a water temperature of 23.98 °C (R^2^ = 0.898). The average rectal temperature on d 30 (*p* = 0.01) and neutral detergent fiber digestibility (*p* < 0.01) increased linearly with increasing water temperature. Additionally, HW_2 reduced serum triiodothyronine, thyroxine, and malondialdehyde (*p* < 0.05), and increased serum total antioxidant capacity (*p* < 0.05) compared with RTW. Compared with HW_2, RTW had unfavorable effects on ruminal propionate, total volatile fatty acids, and cellulase concentrations (*p* < 0.05), and lower relative mRNA expression levels of claudin-4 (*p* < 0.01), occludin (*p* = 0.02), and zonula occludens-1 (*p* = 0.01) in the ruminal epithelium. Furthermore, RTW had a higher abundance of *Prevotella* (*p* = 0.04), *Succinivibrionaceae_UCG-002* (*p* = 0.03), and *Lachnospiraceae_UCG-004* (*p* = 0.03), and a lower abundance of *Bifidobacteriaceae* (*p* < 0.01) and *Marinilabiliaceae* (*p* = 0.05) in rumen compared to HW_2. Taken together, heated drinking water in cold climates could positively impact the growth performance, nutrient digestibility, antioxidant capacity, and rumen fermentation function of beef cattle. The optimal water temperature for maximizing ADG was calculated to be 23.98 °C under our conditions. Ruminal propionate and its producing bacteria including *Prevotella, Succinivibrionaceae,* and *Lachnospiraceae* might be important regulators of rumen fermentation of beef cattle drinking RTW under cold conditions.

## 1. Introduction

Sustainability and animal welfare are emerging as two key development goals in the global livestock industry [1]. The cold climate in winter could induce changes in the endocrine homeostasis and metabolism of beef cattle, resulting in cold stress responses [2]. Cold stress induces hormonal changes to adapt to external pressure, including the activation of the hypothalamic–pituitary–thyroid axis, resulting in an elevation of serum T3 and T4 levels [3]. Wang et al. [4]. demonstrated that prolonged cold stress increases the levels of serum T3 and T4 of cattle. Additionally, cold stress triggers oxidative stress, decreases immune and anti-inflammatory capabilities [5,6], and leads to modifications in rumen fermentation parameters of ruminants [7]. Consequently, these effects may impede ruminant digestive functions and ultimately compromise growth and health [4]. The occurrence of extreme cold events has increased in recent years [8], which necessitates greater attention to winter husbandry management for the health and welfare of livestock.

Water is an important but often overlooked nutrient factor that highly affects the rumen temperature of beef cattle, especially in cold climates [9]. Petersen et al. [10] reported that cows with access to warm water (31.1 ± 1.3 °C) had a significantly lower proportion of ruminal temperature, dropping below 38 °C compared to cold water (8.2 ± 0.4 °C), with rumen temperatures ranging from 34.5 to 40.6 °C and 31.6 to 40.8 °C, respectively. Decreased rumen temperature might inhibit microbial activity [11] and reduce adhesion to fibrous substrates [12]. It has been demonstrated that rumen temperature is highly positively correlated with core body temperature [13], and a decrease in body temperature could lead to oxidative stress and decreased immune function [6,14], which adversely affects the health and welfare of ruminants.

Heated drinking water for beef cattle plays a positive role in mitigating adverse effects of cold weather conditions. Research found that heated drinking water in winter could significantly reduce the duration of the rumen temperature below 37 or 39 °C, and increase the average daily gain (ADG) of Charolais cattle [15]. Our laboratory also reached consistent conclusions that heated drinking water in winter could increase the drinking frequency and the body surface temperature near the rumen of beef cattle [16], as well as improve ADG and economic efficiency [17]. Therefore, heating the water in winter could be an effective strategy to improve rumen stability, feed efficiency, and growth performance of beef cattle. However, the higher water temperature does not always result in higher performance [10], and heating drinking water increases the equipment investment and electricity costs on the farm. In addition, higher water temperatures could also easily lead to algae blooms, causing eutrophication [18], which has adverse effects on the health of beef cattle [19]. However, limited research focuses on the appropriate water temperature range and its comprehensive effects on antioxidant stress capacity, rumen fermentation function, and the health of beef cattle in cold climates. Only a preliminary in vitro experiment conducted by the author suggested that lowering the in vitro incubation temperature had an adverse effect on propionate production [20]. The lack of the aforementioned information hinders a clear understanding of the rational utilization and underlying mechanisms of heated drinking water in beef cattle farming practices, making it difficult to seek more optimized alternative solutions. This study was undertaken to evaluate the effects of heated drinking water on the growth performance, antioxidant capacity, health level, and rumen fermentation function of beef cattle. The primary objective was to determine the suitable water heating temperature for beef cattle in cold conditions and research the theoretical basis for enhancing growth and health through the consumption of heated water.

## 2. Materials and Methods

The experiment was approved by the Animal Care and Use Committee of China Agricultural University (approval number AW71012022-1-1). And the experiments were conducted at Lianwang Animal Husbandry (Shangqiu, Henan, China).

### 2.1. Animals, Design, and Management

A total of 40 beef cattle (640 ± 19.2 kg) aged 22 to 23 months were randomly divided into 5 treatments according to body weight (BW), and each treatment contained 8 cattle (3.8 m^2^ per cattle). The differences among treatments were only drinking water temperature, including a room-temperature water (RTW) group and 4 different heated water (HW) groups, named HW_1, HW_2, HW_3, HW_4, and the target water temperatures for 5 treatments were 4.00 °C, 8.00 °C, 16.00 °C, 22.00 °C, and 28.00 °C. The water for each treatment was provided by the same type of automatic electric heating water tank (length × width × height = 1.50 × 0.60 × 0.65 m; Kangkaijie Agricultural Technology Co., Ltd., Beijing, China) equipped with a temperature sensor to maintain the drinking water temperature required for the experiment. The feeding regimen was the same for all treatments and the diet is shown in Appendix A. The feed was administered in the form of total mixed ration (TMR), administered ad libitum and delivered two times a day at 7:00 a.m. and 15:00 p.m. The nutrient content of TMR was formulated to meet the growth needs of the animals, as required by the Nutrient Requirement Council (2016) [21].

The experiment lasted for 60 d. The individual BW was recorded for 3 consecutive days before morning feeding at the beginning, middle, and end of the experiment, and the ADG was calculated based on the difference. According to observations of the peak water intake periods in beef cattle prior to the experiment, water temperature was measured and recorded at 8:00, 12:00, 16:00, and 18:00 daily and rectal temperature was recorded at the same timestamp on d 0, d 30, and d 60 using a handheld temperature meter (Testo 635, Testo International Trading Co., Ltd., Shanghai, China). The environment temperature and relative humidity were continuously recorded every 0.5 h in a vertical space of 1.7 m above the ground by a temperature and humidity recorder (Apresys 179A-TH, Apresys Optoelectronics Co., Ltd., Shanghai, China). THI was calculated using the formula developed by the M’Hamdi et al. [22]: THI = (1.8 × ET + 32) − [(0.55 − 0.0055 × RH) × (1.8 × ET − 26)].

### 2.2. Sample Collecion

Three days before the end of the experiment, 300 g of TMR samples from each treatment and 300 g of fresh feces samples from each cattle were collected every day and mixed thoroughly; and a subsample of 300 g was taken out, dried in a forced-air oven at 55 °C for 72 h, ground in a Wiley mill (Arthur H. Thomas Co., Philadelphia, PA, USA) to pass a 2 mm sieve, and then pooled within the diversion for further determination of nutrient composition. 

Based on the ADG of each treatment of beef cattle, we selected the lowest (RTW) and highest (HW_2) ADG treatments for further comparison. Cattles with BW close to the average within the RTW and HW_2 groups (n = 6) were selected for blood, rumen fluid, rumen epithelial tissue, and meat quality samples. Blood samples (approximately 5 mL) were collected from the tail vein before feeding in the morning on d 61, were collected into heparinized tubes and immediately centrifuged at 3000× *g* for 15 min at 4 °C to obtain serum samples, and kept at −80 °C prior to analysis. Rumen fluid was aspirated using an esophageal stomach tube 2 h after the morning feeding on d 61. The initial 200 mL rumen fluid was discarded and the remainder was filtered through four layers of sterile gauze and then divided into three 2.0 mL Self-Standing Sample Vials (NEST Biotechnology Co., Ltd., Wuxi, China) and were stored in liquid nitrogen for further analysis. On d 62, the cattle collected were transported to a commercial slaughterhouse and humanely slaughtered. The pre-slaughter weight, hot carcass weight, net meat weight, and bone weight were recorded, and the longissimus dorsi muscle was collected for the meat quality test. Rumen tissue was quickly excised from the ventral blind sac and washed with sterile 0.01 M PBS (pH 6.8) after slaughter. The rumen papillae were scraped to remove attached feed particles and rinsed three times to remove non-adherent bacteria. Epithelial cells for RNA extraction were separated from the muscle layer and stored in 2.0 mL Self-Standing Sample Vials (NEST Biotechnology Co., Ltd., Wuxi, China). The samples were then stored in liquid nitrogen for further analysis. Before the sample collection was completed, the drinking water temperature and feeding management of each treatment remained unchanged.

### 2.3. Chemical Analysis

#### 2.3.1. Digestibility Analysis

The dry matter (DM), crude ash (Ash), crude protein (CP), and ether extract (EE) content of TMR and fecal samples were determined according to the methods described by AOAC (1990) [23]. Organic matter (OM) content was calculated using the formula “1-Ash”. Neutral detergent fiber (NDF) and acid detergent fiber (ADF) were measured using the methods described by van Soest et al. [24]. Apparent digestibility of nutrients was determined using the acid-insoluble ash method described by VanKeulen and Young (1977) [25]. The calculation formula is as follows:D = [1 − (Ad × Nf)/(Af × Nd)] × 100
where Ad (g/kg) and Af (g/kg) represent the acid-insoluble ash in the diet and feces, respectively; Nd (g/kg) and Nf (g/kg) represent the nutrient content in the diet and feces, respectively.

#### 2.3.2. Serum Index

Cortisol (Cor), aldosterone (ALD), triiodothyronine (T3), thyroxine (T4), total antioxidant capacity (T-AOC), superoxide dismutase (SOD), glutathione peroxidase (GSH-Px), and malondialdehyde (MDA) were analyzed by a CLS880 fully automatic biochemical analyzer (Zecen Biotech, Jiangyin, China).

#### 2.3.3. Rumen Fermentation Parameters

The pH of the rumen fluid was measured immediately using a digital-type pH meter (PHS-3C; Yueping Scientific Instrument Co., Ltd., Shanghai, China). The concentration of ammonia nitrogen was determined using the method described by Weatherburn et al. [26] and measured using a spectrophotometer (UV-1700, Shimadzu Corporation, Kyoto, Japan). The concentration of volatile fatty acids (VFA) in the rumen fluid was quantified using high-performance gas chromatography (GC-8600; Beifen Tianpu Instrument Technology Co., Ltd., Beijing, China).

### 2.4. Meat Quality

The pH of the longissimus dorsi muscle was measured 45 min after slaughter using a pH meter (Cyberscan PH310; EUTECH, Singapore). The lightness (L*), redness (a*), and yellowness (b*) values were measured using a colorimeter (Shanghai Precision Scientific Instruments, Shanghai, China). Each sample was measured three times for the same part and the mean value was calculated. The regularly shaped longissimus dorsi muscle was placed in sealed bags, and the contact between the meat samples and the inner wall of the bags was minimized. Then, the samples were suspended in a refrigerator at 4 °C and removed after 24 h. The calculated formation of drip loss was as follows:Drip loss = [ (Initial weight − Final weight)]/Initial weight] × 100%(1)

### 2.5. DNA Extraction, PCR Amplification, and MiSeq Sequencing

#### 2.5.1. Rumen Epithelial Tissue Sample Measurement of Relative Expression of RNA

The procedure for extracting total RNA from the rumen epithelial tissue samples was described by Chomczynski and Sacchi [27]. Briefly, RNA concentration was quantified using the Nanodrop ND-1000UV-Vis spectrophotometer (Thermo Fisher Scientific, Madison, WI, USA). Equal amounts of RNA samples were subjected to electrophoresis on a 1.4% agarose-formaldehyde gel to verify integrity. According to the manufacturer’s instructions, total RNA (1 µg) was reverse-transcribed using the PrimeScript RT kit with gDNA Eraser (Takara Bio Inc., Beijing, China). Primer sets were designed to identify and amplify conserved nucleotide sequences encoding bovine TJ protein and cytokines. The cDNA sequences were identified using the Basic Local Alignment Search Tool (BLAST, National Center for Biotechnology Information, Bethesda, MD, USA) and primers were designed using Primer 5.0 (Whitehead Institute, Cambridge, MD, USA). All primers were synthesized by Genenode Biotechnologies (Beijing, China). Real-time quantitative PCR was performed on an ABI 7500 system (Applied Biosystems, Foster, CA, USA) using SYBR Green fluorescence detection to quantify the target genes and β-actin as a reference gene. All measurements were performed in triplicate. Reverse-transcription negative controls and no-template controls were included as negative controls. The relative abundance of mRNA for each gene of interest was normalized to the mRNA level of the reference gene β-actin, and data were analyzed using the 2^−ΔΔCT^ method. Primer sequences and amplicon sizes for all genes are listed in Appendix A.

#### 2.5.2. DNA Extraction, High-Throughput Sequencing, and Data Processing

The DNA extraction, PCR amplification, and MiSeq sequencing of 12 rumen fluid samples were outsourced to the Allwegene company located in Beijing, China. Refer to the study by He et al. [20] for detailed detection and analysis steps. In summary, DNA was extracted from the rumen fluid samples using the Bacterial DNA Kit (Omega Bio-Tek Inc., Norcross, GA, USA). The V3-V4 region of the bacterial gene was amplified from the extracted DNA using the barcode primers 338F (5-ACTCCTACGGGAGGCAGCAG-3) and 806R (5-GGACTACHVGGGTWTCTAAT-3) rRNA. The amplified PCR products were analyzed by 1% agarose gel electrophoresis and purified using the Agencourt AMPure XP kit (Becker Coulter, Inc., Brea, CA, USA). The purified amplicons were pooled equimolarly and subjected to paired-end sequencing on the Illumina MiSeq platform (Illumina, Inc., San Diego, CA, USA).

QIIME (version 1.17) was used for demultiplexing and quality filtering of the raw fastq files. The sequences overlapping more than 10 bp were assembled using UPARSE. Operational taxonomic units (OTUs) were classified using the RDP classifier and the RDP OTU database (https://www.arb-silva.de/, accessed on 10 June 2023) with a similarity threshold of 97% and a confidence level of 80%. Chimeric sequences were identified and removed using UCHIME. The relative abundance of bacteria was expressed as percentages based on taxonomic analysis. A representative sequence was selected from each OTU based on its abundance. OTUs were used to generate rarefaction curves and calculate alpha diversity indices, including abundance-based coverage estimator (ACE), Chao1, Shannon, and Simpson estimators. To visualize changes in the microbial population structure, jackknifed beta diversity was analyzed through Principal co-ordinate analysis (PCoA) using the UnscramblerX program (CAMO Software Inc. in Woodbridge, NJ, USA).

### 2.6. Statistical Analysis

The data of ADG, nutrient digestibility, and rectal temperature were analyzed using the MIXED procedure of SAS 9.2 (SAS Institute Inc., Cary, NC, USA) with beef cattle as the measurement unit. The linear and quadratic effects caused by the different drinking water temperature were calculated by polynomial contrasts. The GLM program was used for one-way ANOVA and the Student’s *t*-test method was used for comparison of the data of serum, meat quality, and rumen fermentation parameters.

A linear regression model, ADG = a × (drinking water temperature)^2^ + b × (drinking water temperature) + c, was used to evaluate the optimal ratio of drinking water temperature to the ADG of fattening cattle, where a, b, and c are constants. The first derivative was set to zero to obtain the drinking water temperature corresponding to the maximum ADG.

Linear discriminant analysis of effect size (LEfSe) was used with the Kruskal–Wallis rank sum test to analyze the difference in the abundance of the microbiota in feces. The linear discriminant analysis (LDA) scores (threshold = ≥2.0) were used to indicate the size of the effect. Significant differences between treatments were declared at *p* value ≤ 0.05. Differences of 0.05 < *p* value ≤ 0.10 were considered a tendency.

## 3. Results

### 3.1. Climate Conditions and Water Temperature

Throughout the experimental period, the average (± SD) temperature, relative humidity, and THI of the environment were 2.15 ± 6.05 °C, 72.6 ± 3.85%, and 39.1 ± 9.46, respectively (Table 1). The highest and lowest environmental temperatures were 17.1 °C and −12.4 °C, respectively. The average water temperature for RTW, HW_1, HW_2, HW_3, and HW_4 treatments were 4.39 ± 2.546 °C, 10.6 ± 1.29 °C, 18.6 ± 1.52 °C, 26.3 ± 1.70 °C, and 32.5 ± 2.62 °C, respectively. In addition, the daily average ambient temperature and drinking water temperature changes are shown in Appendix A, respectively.

### 3.2. Growth Performance

The BW of the cattle in each treatment showed no significant differences at d 0, d 30, and d 60 (Table 2). The ADG exhibited a linear or quadratic increase with rising water temperature during d 0 to 30 (*p* = 0.04) and d 31 to 60 (*p* = 0.03). Moreover, a significant linear increase (*p* = 0.02) and a tendency toward a quadratic change (*p* = 0.09) in ADG was observed over d 0 to 60.

Figure 1 shows the quadratic model of ADG of beef cattle plotted against the drinking water temperature. The quadratic curve equation was y = −0.0002627 (drinking water temperature)^2^ + 0.0126 (drinking water temperature) + 1.0403, and R² was equal to 0.8981. When the drinking water temperature was equal to 23.98 °C, the ADG reached the highest point of 1.1911 kg/d.

### 3.3. Rectal Temperature

The results of rectal temperature (Table 3) showed that with the increase in drinking water temperature, the average rectal temperature of beef cattle increased linearly on d 30 (*p* = 0.01), and tended to linearly increase at 6:00 on d 30 (*p* = 0.05) and 6:00 (*p* = 0.06) and 8:00 (*p* = 0.09) on d 60.

### 3.4. Nutrient Utilization

With the increase in drinking water temperature (Table 4), the digestibility of NDF showed a significant linear (*p* < 0.01) and quadratic (*p* = 0.04) increase, while the ADF displayed a linear increasing trend (*p* = 0.06). In addition, no significant effect was observed in the digestibility of DM, CP, OM, and EE with the increase in drinking water temperature.

### 3.5. Serum Antistress and Antioxidant Parameters

Compared to RTW, HW_2 showed significant decreases in serum T3 (*p* = 0.01), T4 (*p* < 0.01), and MDA (*p* = 0.02), while serum T-AOC (*p* < 0.01) exhibited a significant increase (Figure 2).

### 3.6. Rumen Fermentation Parameters

The results of rumen fermentation parameters (Table 5) showed that the concentrations of propionate (*p* = 0.01) and T-VFA (*p* = 0.02) in the rumen fluid of HW_2 significantly increased, while its A/P ratio showed a decreasing trend (*p* = 0.06) compared to RTW.

### 3.7. Digestive Enzymes of Rumen Fluid

As shown in Figure 3, compared to RTW, the concentration of cellulase in the rumen fluid of HW_2 significantly increased (*p* = 0.03), and the xylanase concentration showed an increasing trend (*p* = 0.06).

### 3.8. Ruminal Epithelial Barrier mRNA Expression

For ruminal epithelial barrier function (Figure 4), HW_2 significantly upregulated the relative mRNA expression levels of Claudin-4 (*p* < 0.01), Occludin (*p* = 0.02), and ZO-1 (*p* = 0.01) in the ruminal epithelium compared to RTW. However, no significant difference was observed between the two treatments in terms of the relative mRNA expression levels of MCT1, MCT4, and SGLT1.

### 3.9. Slaughter Performance and Meat Quality

Compared to RTW, HW_2 showed an increasing trend in net meat percentage (*p* = 0.08) and bone weight (*p* = 0.08) of beef cattle (Table 6), while its drip loss of the longissimus dorsi muscle showed a decreasing trend (*p* = 0.08).

### 3.10. Bacterial Sequencing and α-Diversity

Bacterial sequencing and examination were performed on 12 rumen fluid samples of beef cattle, yielding a total of 461,929 optimized sequences with an average length of 419 bp (Appendix A). After random subsampling based on the minimum value of sample sequences, a total of 2132 OTUs were discovered, which were classified into 18 phyla, 34 classes, 67 orders, 118 families, 251 genera, and 525 species based on comparison with the Silva database.

In terms of alpha diversity, no significant differences were observed between the RTW and HW_2 groups in Sobs, Shannon, Simpson, Chao, Coverage, and Ace indices (Figure 5).

### 3.11. Bacterial Composition and β-Diversity Analysis

The Venn analysis revealed 1681 shared OTUs, as well as 249 unique OTUs of RTW and 202 unique OTUs of HW_2 (Figure 6A). And there was no significant difference in principal coordinate analysis between RTW and HW_2 at the OTU level (PCoA: R = 0.0333, *p* = 0.326) (Figure 6B). The microbial composition was visualized through bar charts and heatmaps at the phylum (Figure 6C) and genus (Figure 6D) levels. The top three microbial groups at the phylum level in RTW were *Bacteroidota* (58.67%), *Firmicutes* (39.17%), and *Actinobacteriota* (0.46%). The top five microbial groups at the genus level in RTW were *Prevotella* (28.38%), *Prevotellaceae_UCG-001* (8.13%), *Succiniclasticum* (7.85%), *NK4A214_group* (6.21%), and *Rikenellaceae_RC9_gut_group* (6.12%). 

Additionally, the main differences in composition of bacteria between RTW and HW_2 groups are shown in Figure 7. Compared to HW_2, the RTW had a lower relative abundance of *Actinobacteriota* (*p* < 0.01) at the phylum level (Figure 7A), and it had a higher abundance of *Marvinbryantia* (*p* = 0.04), *Prevotella* (*p* = 0.04), *Anaerovibrio* (*p* < 0.01), *Succinivibrionaceae_UCG-002* (*p* = 0.03), and *Lachnospiraceae_UCG-004 (p =* 0.03) and a lower abundance of *Bifidobacteriaceae* (*p* < 0.01), *Atopobium* (*p* = 0.03), and *Marinilabiliaceae* (*p* = 0.05) at the genus level (Figure 7B). Furthermore, LEfSe (Figure 7D) revealed some other different bacteria. Compared to HW_2, the relative abundance of *Oscillospirales* was significantly increased and the relative abundance of *Actinobacteria, Clostridium_methylpentosum_group,* and *Erysipelotrichaceae_UCG-008* was significantly decreased in RTW (LDA > 2.0, *p* < 0.05). 

### 3.12. Spearman Correlation Analysis of the Top 50 Bacteria Genera with Other Parameters

According to the Spearman correlation analysis (Figure 8), *Bacteroidales_UCG-001* was significantly positively correlated with the relative mRNA expression levels of ZO-1, rumen propionate, and xylanase (r > 0.64, *p* < 0.05). ADG during d 0 to 60 was positively correlated with *Bifidobacteriaceae* and *Anaeroplasma* (r > 0.60, *p* < 0.05) while negatively correlated with *Prevotellaceae*_*UCG*-*001* and *Bacteroidales* (r < −0.59, *p* < 0.05). The rumen cellulase, xylanase, and propionate were positively correlated with *Prevotella*, *Prevotellaceae*_*UCG*-*003*, and *Sphaerochaeta* (r > 0.61, *p* < 0.05) and were negatively correlated with *Lachnospiraceae*, *Lachnospiraceae*_*AC2044*_*group*, and *Ruminococcus* (r < −0.58, *p* < 0.05). 

## 4. Discussion

Exposure to a cold climate could induce metabolic changes in beef cattle, leading to decreased growth performance and production efficiency [28]. Previous studies have defined mild, moderate, and severe cold stress temperatures for cattle as 0 to −6.7 °C, −7.2 to −13.9 °C, or <−13.9 °C, respectively [29]. In our experiment, the daily average environmental temperature ranged from 0 to −6.7 °C for 26 days (mainly concentrated on d 31 to 60, including the sampling period), with one day (d 56, ET = −7.72 °C) falling into the −7.2 to −13.9 °C category, which means that 45% of the experimental period the cattle were under mild to moderate cold stress. However, the ADG and the digestibility of NDF and ADF were increased with increasing water temperature, which suggested that heated drinking water could mitigate the negative impact of cold stress on cattle. Our findings are partially consistent with previous studies conducted in our laboratory, although the improvement in ADG was lower compared to previous studies [16]. In addition, Grossi et al. (2021) [15] also found a significant increase (+3.5%) in the ADG of cattle drinking water heated to 25 °C compared to room-temperature water in winter. The lower ADG of RTW might result from its lower nutrient digestibility. Consistent with our results, previous studies showed that cold environments led to decreased digestibility of DM, NDF, and ADF in growing cattle [4], calves [2,5], lambs [30], and mice [31]. Cold water intake significantly decreases rumen temperature in cattle [9,13,15], which might inhibit rumen microbial activity [11], diversity [32], and adhesion to fibrous substrates [12], leading to a change in overall rumen fermentation patterns and a decrease in nutrient digestibility [10,32]. Specifically, changes in temperature might impact the digestion and degradation of fiber [33]. In addition, cellulase and xylanase could promote NDF digestibility [34,35] by improving rumen microbial colonization [36] and synergism [37]. The decrease in observed rumen cellulase and xylanase of RTW might also contribute to lower NDF and ADF digestibility. Furthermore, cold stimulation could enhance gastric and intestinal contractions and peristalsis, leading to a shorter retention time of nutrients in the intestines and incomplete digestion and absorption of the diet, which could also result in decreased nutrient digestibility [38]. Heated drinking water helped maintain a relatively stable rumen temperature [13], which in turn reduces energy losses and preserves rumen fermentation function [11], which might be the primary reason for enhancing cold resistance and growth performance in beef cattle. In this experiment, we housed cattle for each treatment in one pen to minimize the fluctuations in water temperature within each treatment, which was beneficial for controlling the experimental variables. However, this approach might have certain limitations in interpreting the results. Nevertheless, the indicators we focused on support the observations and statistics of individual beef cattle as experimental units. The present study provided a certain reference for future research. Subsequent investigations could delve deeper into the impact of water temperature on cattle growth and feed efficiency by incorporating more pens as experimental units.

Studies have demonstrated that cold stress induces stress responses in ruminants, resulting in elevated concentrations of serum glucocorticoids, triiodothyronine, and thyroxine [39]. In our study, the RTW increased the concentrations of serum T3 and T4, which could contribute to the promotion of gluconeogenesis and hepatic glycogen synthesis, enhance thermogenesis, and improve the resistance of the organism to low-temperature environments [40]. Furthermore, cold environments cause trembling in animals, leading to increased oxygen consumption [41], and decreased oxygen concentration in the body typically results in increased oxidative stress [42]. We found a significant increase in serum MDA concentration and a decrease in T-AOC in cattle of RTW. This might be due to the RTW intensifying the extent and recovery time of rumen temperature reduction in cattle [15], and rumen temperature is positively correlated with the body core temperature [43,44]. Although this study did not measure changes in rumen temperature, we found that with increasing water temperature, the rectal temperature of beef cattle significantly increased in a linear fashion on day 30 and in the morning of day 60, which indicated that consuming heated water mitigated the decline in the rectal temperature of beef cattle under cold conditions. This intervention also enhanced the anti-stress and antioxidant capacity of the cattle, thereby helping to maintain homeostasis and health in challenging low-temperature conditions.

The rumen epithelium serves as a crucial tissue for the interaction between the host and microbiota, as well as the absorption and utilization of nutrients in ruminants [45]. Our study revealed that the RTW group had a lower relative gene expression of Claudin-4, Occludin, and ZO-1 in the rumen epithelium, which indicated that cold water intake had a detrimental effect on the rumen epithelial barrier function in cold environments [45]. Previous studies found that nutritional stress during cold seasons led to a decrease in the gene expression of Claudin-4, ZO-1, and SGLT1 in rumen epithelium of Tibetan antelopes [46]. An increased intestinal barrier permeability and decreased expression of tight junction proteins were also found in ruminants with low feed intake [47,48,49]. Although feed intake was not measured in our study, previous research demonstrated that drinking cold water significantly reduced dry matter intake [15] and water intake [10] in beef cattle. Moreover, the decrease in rumen temperature caused by cold water might activate transient receptor potential channels as temperature sensors, leading to intracellular Ca^2+^ release and activation of TAK1, subsequently activating the NFκB signaling pathway [50], which may induce the expression of pro-inflammatory factors and suppression of the antioxidant system [51,52]. In our study, lower antioxidant and anti-stress abilities were found in the RTW group, which might contribute to impaired rumen epithelial barrier function.

Drinking cold water during the winter imposes an additional energy demand on beef cattle [15]. In our study, decreased rumen propionate and T-VFA concentrations were observed in RTW. Propionate serves as a key substrate for gluconeogenesis, activating the expression of gluconeogenic genes to maintain energy homeostasis in the body [53]. Previous studies demonstrated a significant reduction in rumen propionate of Korean cattle [2] or sheep [33] subjected to mild cold stress. The reduction in propionate might be attributed to the increased demand for propionate absorption in beef cattle under cold conditions to support gluconeogenesis, providing the body with more energy to withstand the cold [33]. The concentration of propionate and other VFA in the rumen was a dynamic balance between microbial fermentation production and utilization by the host. We attribute the decrease in rumen propionate and total VFA concentrations in the RTW group to increased energy demands in beef cattle under cold conditions, reduced concentrations of digestive enzymes, and inhibited NDF digestion, resulting in an overall decline in VFA production and accumulation.

No significant statistical differences were observed in alpha diversity and principal coordinates analysis between the RTW and HW_2 treatments, indicating that water temperature had less effect on the diversity, abundance, and bacterial community structure of the rumen microbiota under the experimental conditions of our study. A high abundance of *Actinobacteriota* was observed in the HW_2 group, which contributed to the improvement in lignocellulose residue degradation [54]. At the genus level, *Bifidobacteriaceae*, *Atopobium*, *Marinilabiliaceae*, *Actinobacteria*, and *Erysipelotrichaceae_UCG-008* were dominant in the HW_2 group. *Bifidobacteriaceae* exhibits probiotic or health-promoting effects on the host and could actively improve microbial communities in the rumen under conditions of rumen acidosis [55,56]. We observed a significant positive correlation between *Bifidobacteriaceae* and ADG, indicating that *Bifidobacteriaceae* plays a beneficial role in promoting the health of beef cattle. A higher abundance of *Prevotella*, *Succinivibrionaceae_UCG-002,* and *Lachnospiraceae_UCG-004* was found in RTW. *Prevotella* had the highest abundance at the genus level in our study and belongs to the phylum *Bacteroidetes*, which mainly degrades non-structural carbohydrates and promotes propionate production [57]. *Succinivibrionaceae* could degrade non-fiber carbohydrates in the rumen into succinate and further converted them into propionate [58]. *Lachnospiraceae_UCG-004* belongs to *Lachnospiraceae*, which could degrade plant cellulose and hemicellulose and convert them into short-chain fatty acids for host absorption [59]. In this experiment, *Prevotella* and *Prevotellaceae*_*UCG*-*003* were positively correlated with rumen cellulase, xylanase, and propionate, indicating that these bacteria play an important role in synthesizing VFAs and energy absorption and utilization. Similar to our findings, Cui et al. [60] also reported a significant increase in the abundance of rumen *Prevotella*, *Lachnospiraceae*, and *Ruminococcus* in Tibetan Sheep in cold environments, which contributed to improving energy efficiency and adaptation to cold conditions. The microbial results of our study indicated that drinking cold water increased the relative abundance of propionate-producing bacteria, specifically *Prevotella Succinivibrionaceae* and *Lachnospiraceae*, which could be an adaptive evolution of the rumen microbiota in response to the higher energy demands of beef cattle under the cold season.

## 5. Conclusions

In conclusion, providing heated water in cold climates could alleviate the negative impacts of cold stimulation, and positively impact the growth performance, nutrient digestibility, antioxidant capacity, and rumen fermentation function of beef cattle. Under the conditions of our study, an optimal water temperature of 23.98 °C resulted in the maximum ADG. Additionally, Ruminal propionate and its producing bacteria including *Prevotella*, *Succinivibrionaceae*, and *Lachnospiraceae* changed significantly, which might be important regulators of rumen fermentation, energy balance, and nutrient utilization in beef cattle drinking water at different temperatures under cold conditions.

## Figures and Tables

**Figure 1 antioxidants-12-01492-f001:**
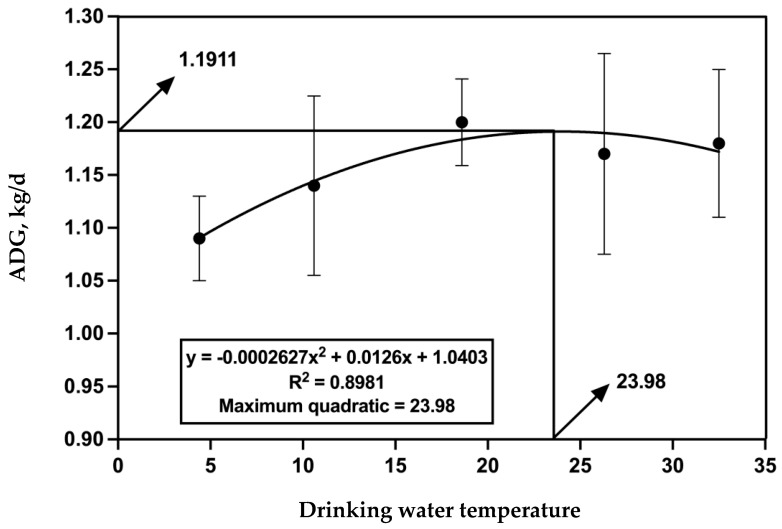
Quadratic model of average daily gain (ADG) of beef cattle plotted against the drinking water temperature. The quadratic curve equation was *y* = −0.0002627 (drinking water temperature)^2^ + 0.0126 (drinking water temperature) + 1.0403, and *R*² was equal to 0.8981. When the drinking water temperature was equal to 23.98 °C, the ADG (1.1911 kg/d) reached the highest point.

**Figure 2 antioxidants-12-01492-f002:**
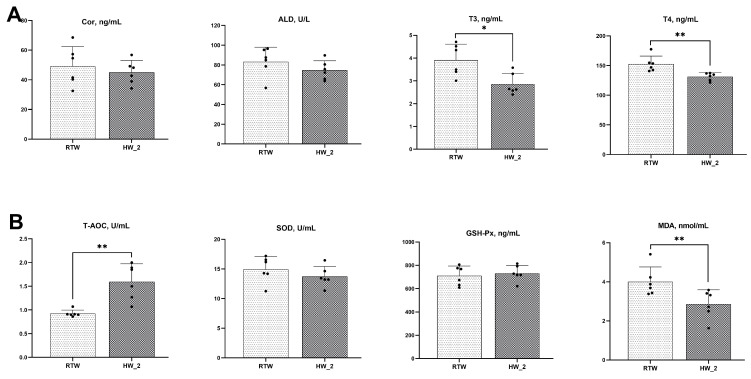
Effect of drinking water temperature on serum antistress (**A**) and antioxidant (**B**) parameters of beef cattle. Cor, Cortisol; ALD, aldosterone; T3, triiodothyronine; T4, thyroxine; T-AOC, total antioxidant capacity; SOD, superoxide dismutase; GSH-Px, glutathione peroxidase; MDA, malondialdehyde. RTW and HW_2 denote drinking room temperature water at 4.39 ± 2.546 °C and heated water at 18.6 ± 1.52 °C, respectively. Bars marked with various asterisks (*) denote the degree of significant differences. *, *p* < 0.05; **, *p* < 0.01. *n* = 6.

**Figure 3 antioxidants-12-01492-f003:**
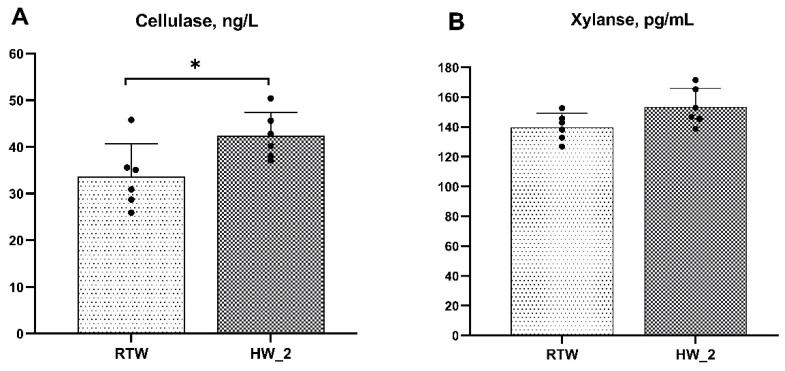
Effect of drinking water temperature on the concentration of cellulase (**A**) and xylanase (**B**) in rumen liquid of beef cattle. RTW and HW_2 denote drinking room temperature water at 4.39 ± 2.546 °C and heated water at 18.6 ± 1.52 °C, respectively. Bars marked with various asterisks (*) denote the degree of significant differences. *, *p* < 0.05. *n* = 6.

**Figure 4 antioxidants-12-01492-f004:**
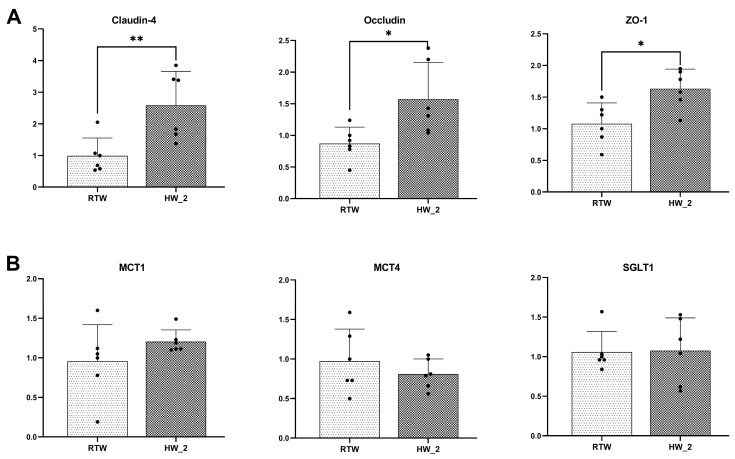
Effect of drinking water temperature on ruminal epithelial barrier function (**A**) and ruminal epithelial transporter (**B**) mRNA expression of beef cattle. ZO-1, zonula occludens-1; MCT1, monocarboxylic acid transporters 1; MCT4, monocarboxylic acid transporters 4; SGLT1, sodium-dependent glucose-linked transporter-1. RTW and HW_2 denote drinking room temperature water at 4.39 ± 2.546 °C and heated water at 18.6 ± 1.52 °C, respectively. Bars marked with various asterisks (*) denote the degree of significant differences. *, *p* < 0.05; **, *p* < 0.01. *n* = 6.

**Figure 5 antioxidants-12-01492-f005:**
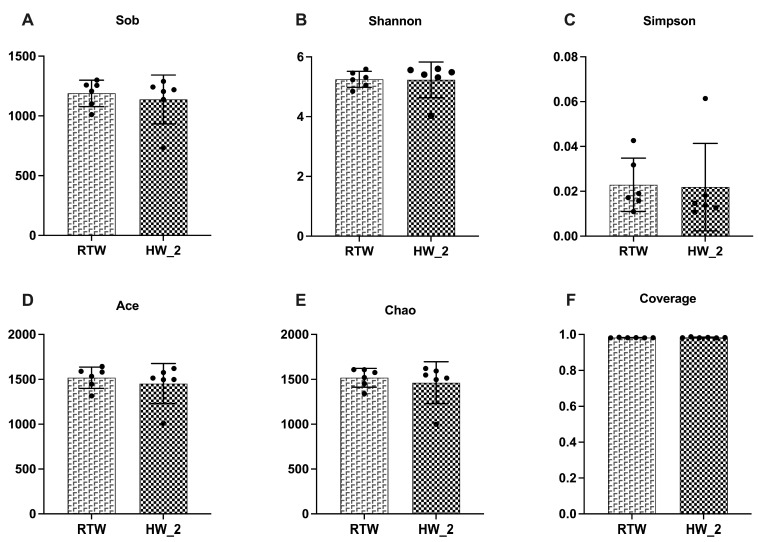
Effect of drinking water temperature on bacterial α-diversity of rumen at OTU level in beef cattle. (**A**) Sobs index; (**B**) Shannon index; (**C**) Simpson index; (**D**) Ace index; (**E**) Chao index; (**F**) Coverage index. RTW and HW_2 denote drinking room temperature water at 4.39 ± 2.546 °C and heated water at 18.6 ± 1.52 °C, respectively. *n* = 6.

**Figure 6 antioxidants-12-01492-f006:**
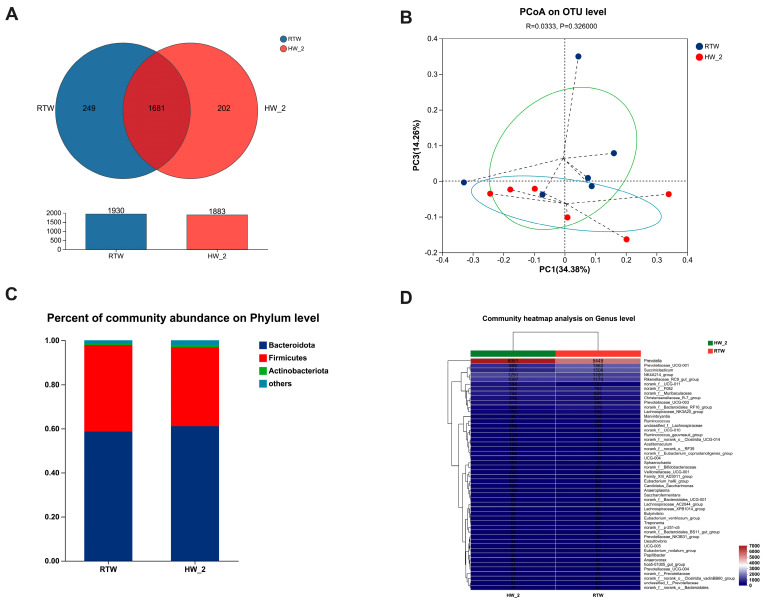
Effect of drinking water temperature on rumen bacterial composition and β−diversity in beef cattle. (**A**) Venn analysis at OUT; (**B**) principal co−ordinate analysis (PCoA) at OTU level; (**C**,**D**) bacterial composition at phylum and genus levels. RTW and HW_2 denote drinking room temperature water at 4.39 ± 2.546 °C and heated water at 18.6 ± 1.52 °C, respectively. *n* = 6.

**Figure 7 antioxidants-12-01492-f007:**
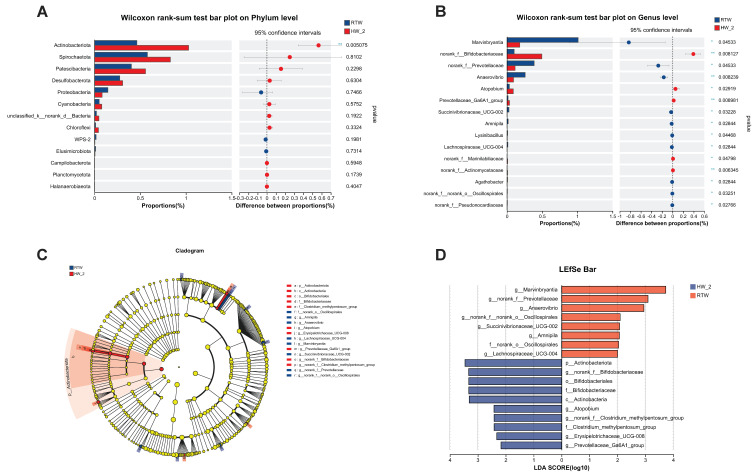
Effect of drinking heated water and room−temperature water on rumen bacterial composition differences of beef cattle. (**A**,**B**) Differences in microorganism at phylum and genus levels. (**C**) Cladogram; (**D**) LDA. LEfSe, linear discriminant analysis of effect size; LDA, linear discriminant analysis. *p* < 0.05 and LDA score > 2.0 are presented. RTW and HW_2 denote drinking room temperature water at 4.39 ± 2.546 °C and heated water at 18.6 ± 1.52 °C, respectively. Bars marked with various asterisks (*) denote the degree of significant differences. *, *p* < 0.05, **, *p* ≤ 0.01, *n* = 6.

**Figure 8 antioxidants-12-01492-f008:**
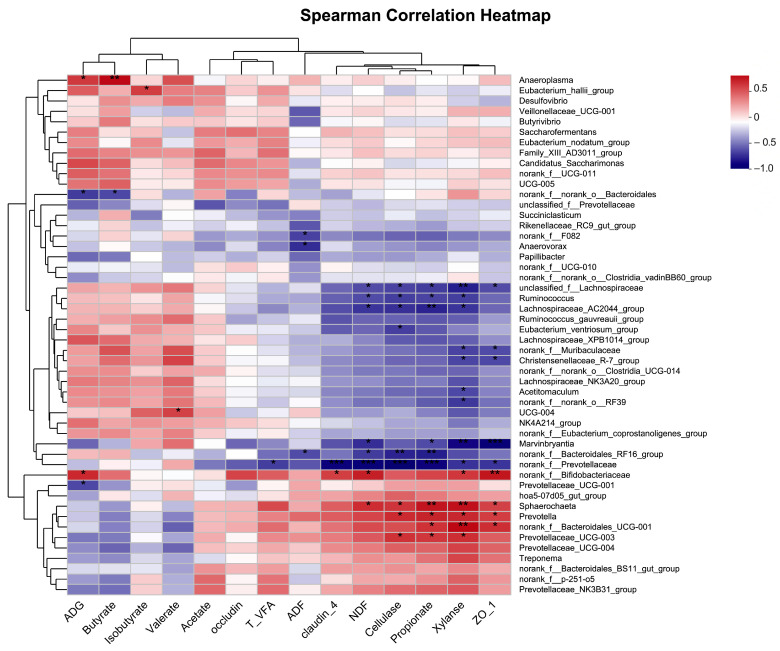
Spearman correlation analyses between the top 50 relatively abundant bacterial genera and other parameters. The X−axis and Y−axis are environmental factors and species, respectively, and the correlation R value and *p* value are obtained by calculation. R values are displayed in different colors in the figure, and the legend on the right is the color interval of different R values. ADG, average daily gain during day 0 to 60; ZO−1, zonula occludens−1; T_VFA: total volatile fatty acids. * *p* ≤ 0.05; ** *p* ≤ 0.01; *** *p* ≤ 0.001.

**Table 1 antioxidants-12-01492-t001:** Environmental parameters and drinking water temperature description during the experiment.

Item	Mean	Minimum	Median	Maximum	SD
Environment parameters					
Temperature, °C	2.15	−7.72	1.85	17.1	6.05
Relative humidity, %	72.6	63.3	72.6	79.9	3.85
THI	39.1	15.3	38.9	62.0	9.46
Water temperature, °C					
RTW	4.39	0.90	3.70	10.9	2.546
HW_1	10.6	7.60	10.7	13.8	1.29
HW_2	18.6	15.4	18.5	22.1	1.52
HW_3	26.3	23.0	26.2	30.4	1.70
HW_4	32.5	27.3	33.3	37.1	2.62

THI, temperature and humidity index; SD, standard deviation; RTW, HW_1, HW_2, HW_3, and HW_4 denote drinking room temperature water at 4.39 ± 2.546 °C and heated drinking water at 10.6 ± 1.29 °C, 18.6 ± 1.52 °C, 26.3 ± 1.70 °C, and 32.5 ± 2.62 °C, respectively.

**Table 2 antioxidants-12-01492-t002:** Effect of drinking water temperature on growth performance of beef cattle.

Item	Treatments					SEM	*p*-Value		
	RTW	HW_1	HW_2	HW_3	HW_4		ANOVA	Linear	Quadratic
BW, kg									
D 0	647	636	641	637	641	7.03	0.84	0.64	0.44
D 30	681	671	678	674	679	7.33	0.88	0.90	0.52
D 60	712	705	713	707	712	7.06	0.89	0.99	0.71
ADG, kg/d									
D 0 to 30	1.15	1.17	1.23	1.23	1.25	0.04	0.28	0.04	0.58
D 31 to 60	1.03	1.11	1.16	1.11	1.11	0.03	0.07	0.11	0.03
D 0 to 60	1.09 b	1.14 ab	1.20 a	1.17 ab	1.18 ab	0.03	0.05	0.02	0.09

SEM, standard error of the mean; BW, body weight; ADG, average daily gain; RTW, HW_1, HW_2, HW_3, and HW_4 denote drinking room temperature water at 4.39 ± 2.546 °C and heated drinking water at 10.6 ± 1.29 °C, 18.6 ± 1.52 °C, 26.3 ± 1.70 °C, and 32.5 ± 2.62 °C, respectively. a,b Values with various superscripts in a row are significant differences (*p* < 0.05). *n* = 8.

**Table 3 antioxidants-12-01492-t003:** Effect of drinking water temperature on rectal temperature of beef cattle (°C).

Time	Treatments					SEM	*p*-Value		
	RTW	HW_1	HW_2	HW_3	HW_4		ANOVA	Linear	Quadratic
D 1									
6:00	38.7	38.6	38.8	38.9	38.8	0.19	0.88	0.49	0.88
8:00	38.9	38.8	38.8	38.8	38.8	0.16	0.99	0.81	0.87
14:00	38.7	38.8	38.5	38.6	38.6	0.19	0.89	0.64	0.92
20:00	38.6	38.8	38.6	39.0	38.7	0.17	0.51	0.55	0.34
Mean	38.7	38.8	38.7	38.8	38.7	0.08	0.80	0.75	0.65
D 30									
6:00	38.5	39.0	38.9	38.9	39.3	0.25	0.25	0.05	0.87
8:00	38.7	38.8	39.1	39.1	39.0	0.22	0.69	0.26	0.46
14:00	38.7	38.7	39.1	38.9	39.0	0.22	0.50	0.24	0.57
20:00	38.5	38.5	38.5	38.8	38.7	0.11	0.32	0.11	0.93
Mean	38.6	38.7	38.9	38.9	39.0	0.11	0.10	0.01	0.46
D 60									
6:00	38.8	38.5	39.1	38.8	39.1	0.18	0.09	0.06	0.53
8:00	38.8	39.0	39.2	39.4	39.2	0.22	0.42	0.09	0.44
14:00	38.9	39.2	38.7	39.0	39.1	0.21	0.63	0.75	0.58
20:00	39.2	38.9	39.0	38.9	38.7	0.20	0.47	0.12	0.98
Mean	38.9	38.9	39.0	39.0	39.0	0.11	0.80	0.29	0.90

RTW, HW_1, HW_2, HW_3, and HW_4 denote drinking room temperature water at 4.39 ± 2.546 °C and heated drinking water at 10.6 ± 1.29 °C, 18.6 ± 1.52 °C, 26.3 ± 1.70 °C, and 32.5 ± 2.62 °C, respectively. *n* = 8.

**Table 4 antioxidants-12-01492-t004:** Effect of drinking water temperature on nutrient digestibility of beef cattle (%).

Item	Treatments					SEM	*p*-Value		
	RTW	HW_1	HW_2	HW_3	HW_4		ANOVA	Linear	Quadratic
DM	73.18	72.62	71.88	72.28	73.27	0.61	0.46	0.94	0.08
CP	72.59	72.92	72.55	71.71	73.78	0.80	0.50	0.65	0.33
NDF	59.36 b	62.08 ab	66.02 a	65.65 a	66.02 a	1.10	<0.01	<0.01	0.04
ADF	45.07	47.04	50.10	49.07	50.09	1.93	0.31	0.06	0.42
OM	59.72	59.52	58.66	60.14	60.86	1.07	0.69	0.40	0.31
EE	75.85	77.99	73.73	71.95	75.95	1.69	0.15	0.28	0.34

SEM, standard error of the mean; DM: dry matter; CP: crude protein; NDF: neutral detergent fiber; ADF: acid detergent fiber; OM, organic matter; EE: ether extract. RTW, HW_1, HW_2, HW_3, and HW_4 denote drinking room temperature water at 4.39 ± 2.546 °C and heated drinking water at 10.6 ± 1.29 °C, 18.6 ± 1.52 °C, 26.3 ± 1.70 °C, and 32.5 ± 2.62 °C, respectively. a,b Values with various superscripts in a row indicate significant differences (*p* < 0.05). *n* = 6.

**Table 5 antioxidants-12-01492-t005:** Effect of drinking water temperature on rumen fermentation parameters of beef cattle.

Item	Treatments		SEM	*p*-Value
	RTW	HW_2		
pH	6.82	6.79	0.05	0.69
NH_3_-N, g/dL	0.03	0.06	0.01	0.08
Acetate, mmol/L	44.88	52.42	2.28	0.1
Propionate, mmol/L	9.37 b	13.35 a	0.83	0.01
Butyrate, mmol/L	3.35	3.99	0.22	0.16
Isobutyrate, mmol/L	0.91	0.96	0.04	0.49
Valerate, mmol/L	1.16	1.25	0.13	0.75
Isovalerate, mmol/L	1.59	1.8	0.19	0.6
A/P	4.79	3.93	0.22	0.06
T-VFA, mmol/L	61.26 b	73.77 a	2.93	0.02

SEM, standard error of the mean; NH_3_-N, ammonia nitrogen; T-VFA, total volatile fatty acid; A/P, the ratio between the content of acetate and propionate. RTW and HW_2 denote drinking room temperature water at 4.39 ± 2.546 °C and heated water at 18.6 ± 1.52 °C, respectively. a,b Values with various superscripts in a row indicate significant differences (*p* < 0.05). *n* = 6.

**Table 6 antioxidants-12-01492-t006:** Effect of drinking water temperature on slaughter performance and meat quality of beef cattle.

Item	Treatments		SEM	*p*-Value
	RTW	HW_2		
Slaughter performance				
BW before slaughter, kg	705	720	10.4	0.33
Hot carcass weight, kg	403	416	4.88	0.09
Net meat weight, kg	331	344	4.82	0.09
Dressing percent, %	57.2	57.8	0.34	0.25
Carcass meat rate, %	82.1	82.7	0.39	0.31
Net meat percentage, %	47.0	47.8	0.30	0.08
Bone weight, kg	63.1	65.9	1.02	0.08
Meat bone ratio	5.25	5.22	0.06	0.77
Meat quality				
pH	6.85	6.82	0.04	0.51
L*	34.7	34.2	0.77	0.67
a*	11.6	11.7	0.89	0.93
b*	7.12	7.85	0.57	0.38
Drip loss,%	10.0	9.34	0.25	0.08
Lion-eye area (cm^2^)	165	168	2.43	0.50

SEM, standard error of the mean; RTW and HW_2 denote drinking room temperature water at 4.39 ± 2.546 °C and heated water at 18.6 ± 1.52 °C, respectively. L*, lightness; a*, redness; b*, yellowness. *n* = 6.

## Data Availability

The original data of this study are included in the article and further information is available upon reasonable request to the corresponding author.

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
