# Peer review of "Heating Drinking Water in Cold Season Improves Growth Performance via Enhancing Antioxidant Capacity and Rumen Fermentation Function of Beef Cattle"

_antioxidants, 2023, doi:10.3390/antiox12081492_

Round 1
Reviewer 1 Report
The manuscript antioxidants-2494967 aimed to study how the temperature of drinking water during wither affects the growth performance, antioxidant capacity, antistress hormones and fermentation in beef cattle. The study is quite interesting and provides insights into homeostatic mechanisms develop in rumen microenvironment to cope with low temperatures.
Although various parameters were studied, in the introduction, the reader in not introduced to the antistress hormones, redox status and others, nor how they relate to rumen health and fermentation. The first characterization of these hormones as antistress hormones is given in Results section. Therefore, there is no complete connection between the Introduction and the objectives of this study with the studied parameters. Some information and references could be used in the Introduction instead of Discussion.
Some minor comments for revision are provided below.
Line 59: Immune damage: what do you mean by this?
Lines 61-62: please rephrase the sentence.
Lines 82-84: The meaning of this sentence is too indefinite to represent the objective of the study.
Line 89: How old were the cattle? Please define it.
Line 232: Delete the ‘And’ at the beginning of the sentence.
Lines 256: Replace "According to Table 2" from the beginning of the sentence at the end or in parentheses.
Lines 292, 310, 338: Please revise ‘were significant differences’.
Lines 294-296: English language editing required.
Line 551: 29 °C or 21?
Discussion: You wrote different temperatures in Lines 404-407 compared to Table 1. Please clarify it.
Please see the Comments and Suggestions for Authors.
Reviewer 2 Report
This manuscript evaluates the effect of drinking water temperature on rumen parameters and animal growth. This is a very interesting paper. It is very interesting that water temperature affected cellulolytic bacteria and NDF digestion (quite dramatically), but not other nutrients.
Specific comments:
L48 - change to the 'The occurence of extreme cold events has increased'
L94 should go in Results. The target water temperature should go here in Methods
L101 - change to 'TMR was formulated to meet'
L104 - change to 'BW was recorded for 3'
L119 - change to 'and a subsample of 300 g was'
L123 - why only these 2 groups
L230 - there were 8 cattle per treatment all in one pen. There is a major problem here. Pen is really the experimental unit and n = 1. By using individual animal data, you really have pseudoreplication. This cannot be corrected post-experiment with data analysis.
L234 - change to 'meat quality'
L235 - this is a linear model - a+b+c - is a linear combination of parameters. It is a polynomial equation, but a linear model. There is no need to use NLIN for this analysis.
Change the phrase 'drinking heated water' to 'heated drinking water' throughout the manuscript
Figure 1 - it looks the the peak ADG is nearer to 25 C, maybe about 23 or 24 C rather than 21 C
L296 - change to 'increased for HW_2'
L314 - not sure what a significant trend is. should just say a trend
L406 - 7.72 C is positive and I think it should be a negative value
L447 - 'that' is used twice
L504-505 - this sentence does not make sense. did you mean to say that heated drinking water could reshape the rumen microbial community?
L515 - I thought 21 C was the optimum water temperature
Table 4 - change title to say 'nutrient digestibility'
Figure 2 - why only compare RTW and HW_2?
Table 5 - mg/kg is an unusual unit of measure for rumen fermentation parameters. I am not sure how to interpret it. I think it should be mmol/ml
Figure 5 - why is this color when other figures are black/white
English language is good. minor grammatical errors found. may need further proofreading for grammatical errors
Round 2
Reviewer 2 Report
The authors have made good improvements to the manuscript and I have no new comments. But I have further addressed the author's response to some of my original comments.
Reply to Author Response
As for why only RTW and HW_2 were used in analysis of antioxidants, etc. Please add your explanation to the manuscript - We selected the lowest and highest ADG treatments for further comparison.
As to the number of experimental units per treatment, I am sorry that I used the incorrect terminology, but does not change the fact that you are using measurement unit as the experimental unit which in this case are not the same unit. It appears that barn in the experimental unit and calf is the measurement unit so you need to run statistics on the mean values for barn which leads to n = 1. Just because another paper was publixhed with bad experimental design does not make it ok. There are some instances where animals in the same pen can still serve as experimental units, but that is based on the biology of the treatments applied. Please provide biological/physiological evidence/argument that animal within barn can serve as the experimental unit in this case.
As to Figure 1, I was not referring to the fact that the temperature for ADG was incorrect (calculation of 21 C). I was suggesting that the graphical representation of this may be incorrect. When I plot your polynomial regression equation in Excel, I get a curve where the maximum ADG looks to be about 20 C.
